# Two-Year-Span Breast Cancer Screening Uptake in Japan after the COVID-19 Pandemic and Its Association with the COVID-19 Vaccination

**DOI:** 10.3390/cancers16091783

**Published:** 2024-05-05

**Authors:** Aminu Kende Abubakar, Yudai Kaneda, Akihiko Ozaki, Hiroaki Saito, Michio Murakami, Daisuke Hori, Kenji Gonda, Masaharu Tsubokura, Takahiro Tabuchi

**Affiliations:** 1Graduate School of Public Health, St. Luke’s International University, Tokyo 104-004, Japan; 2Hokkaido University School of Medicine, Hokkaido 060-8638, Japan; nature271828@gmail.com; 3Breast and Thyroid Center, Jyoban Hospital of Tokiwa Foundation, Fukushima 972-8322, Japan; ozakiakihiko@gmail.com (A.O.); gondake@kind.ocn.ne.jp (K.G.); 4Medical Governance Research Institute, Tokyo 108-0074, Japan; 5Department of Internal Medicine, Soma Central Hospital, Fukushima 976-0011, Japan; h.saito0515@gmail.com; 6Department of Radiation Health Management, Fukushima Medical University, Fukushima 960-1295, Japan; tsubokura_tky@me.com; 7Center for Infectious Disease Education and Research, Osaka University, Osaka 565-0871, Japan; michio@cider.osaka-u.ac.jp; 8Institute of Medicine, University of Tsukuba, Ibaraki 305-8575, Japan; daisuke_hori@md.tsukuba.ac.jp; 9Division of Epidemiology, School of Public Health, Tohoku University Graduate School of Medicine, Miyagi 980-0872, Japan; tabuchitak@gmail.com; 10Cancer Control Center, Osaka International Cancer Institute, Osaka 541-8567, Japan

**Keywords:** breast cancer, cancer screening, COVID-19, vaccination, healthcare, pandemic, disaster, Japan

## Abstract

**Simple Summary:**

Lockdowns and health service disruptions during the COVID-19 pandemic led to concerns about a potential decrease in essential health screenings, even after the pandemic. It is crucial to understand how the pandemic affected women’s participation in breast cancer screenings in Japan and whether getting the COVID-19 vaccine influenced their participation rates. We analyzed data from over 6110 women aged 40 to 74 years from a large online survey conducted in 2021 and 2022. Our findings showed that the number of women getting screened for breast cancer did not decrease after the pandemic. Moreover, women who were vaccinated against COVID-19 were more likely to attend their screenings compared to those who were not vaccinated. This result aligns with the observation that vaccinated individuals tend to be more proactive about their health.

**Abstract:**

There is limited information on whether the COVID-19 pandemic was associated with decreased breast cancer screening uptake and if COVID-19 vaccination was associated with an increase in screening uptake. Our study explored the uptake of breast cancer screening in Japan after the COVID-19 pandemic and assessed its association with the COVID-19 vaccination. We analyzed data from the Japan COVID-19 and Society Internet Survey (JACSIS), a web-based prospective cohort survey, and we included 6110 women without cancer history who were aged 40 to 74 years that participated in the 2012 and 2022 surveys. We examined the regular breast cancer screening uptake before and after the pandemic and employed a multivariable Poisson regression model to seek any association between COVID-19 vaccination and screening uptake. Of 6110, 38.2% regularly participated in screening before the pandemic and 46.9% did so after the pandemic. Individuals unvaccinated due to health reasons (incidence rate ratio (IRR) = 0.47, 95% CI: 0.29–0.77, *p* = 0.003) and for other reasons (IRR = 0.73, 95% CI: 0.62–0.86, *p* < 0.001) were less likely to undergo screening compared to fully vaccinated individuals. There was no long-term decrease in breast cancer screening uptake after the pandemic in Japan. Vaccination was linked to increased uptake, but there was no dose relationship.

## 1. Introduction

Breast cancer is one of the most common cancers among women, with an estimated 2.3 million new cases diagnosed globally in 2020 [1]. The peak ages of onset of breast cancer in Japan are between 45 and 49 years and 65 and 69 years [2]. In Japan, it is estimated that breast cancer accounts for 12% of cancer-related deaths among women [3], underscoring the importance of early diagnosis, treatment, and the established benefits of regular screening programs, particularly mammography [4,5]. However, in contrast to Western countries where 70–80% of women have undergone mammography [6], the screening rate in Japan remained at 47.4% in 2019 [7]. Moreover, within Japan, there are significant disparities in uptakes [8], and non-participation in breast cancer screening programs is related to various personal and external factors, such as low educational level, psychological distress, and a lack of social support [9,10].

Over the past few years, concern regarding the impact of disasters and crises on breast cancer screening has been on the rise, and this debate has been further fueled by the novel coronavirus disease 2019 (COVID-19) pandemic [11]. Few studies had considered the effects of disasters and crises on overall cancer screening programs before this pandemic [12]. Since the onset of the COVID-19 pandemic in November 2019, worldwide breast cancer screening rates have declined, partly due to fear of infection and interruptions in medical services [13,14,15]. Consequently, as calamities and emergencies continue to surge, the need to consider the extrinsic factors affecting participation in breast cancer screening has become increasingly pressing.

Unlike other disasters, the COVID-19 pandemic presented a unique situation where preventive measures such as vaccination were established [16]. As of our analysis in January 2023, despite concerns about potential side effects leading some to avoid vaccination, over 80% of the Japanese population had already completed the initially recommended two-dose vaccination protocol [17,18,19]. Moreover, receiving the COVID-19 vaccine was reported to be associated with more positive attitudes towards preventive measures and a higher willingness to change health behaviors [20], and it is widely recognized that individuals who engage in one form of preventive behavior are often predisposed to participate in others. However, the unique global impact of the COVID-19 pandemic presents a novel context for examining these behaviors. Despite Japan’s high vaccination coverage, its cancer screening rates remain comparatively low [6,7], and this discrepancy highlights the need to explore how the pandemic has affected traditional health behaviors and methods to improve the integration of preventive measures in public health campaigns. Therefore, the extensive vaccination effort might have played a pivotal role in transforming people’s behaviors, potentially serving as a significant factor for breast cancer screening participation. There is yet to be a comprehensive report summarizing findings on breast cancer screening rates. Understanding these dynamics is critical, both in Japan and abroad, where the risks of new infectious disease pandemics, such as avian influenza and mpox, have been noted [21,22,23].

This study aimed to investigate the association between COVID-19 vaccination status and post-pandemic breast cancer screening uptake, while accounting for other sociodemographic, behavioral, and health-related factors.

## 2. Materials and Methods

### 2.1. Settings and Participants

We used the data from the Japan COVID-19 and Society Internet Survey study (JACSIS), which is an ongoing cohort study designed to recruit a ‘nationally representative sample’ to calculate national estimates. The web-based, self-administered survey was distributed by an internet research agency (Rakuten Insight, Inc, Tokyo, Japan) which had approximately 2.3 million registered qualified panelists. More information regarding the survey and questionnaire is available on the study website [24].

In this study, we included women with no history of cancer who were aged 40 and 74 at the time of the 2021 survey, which was conducted in September–October 2021, and also who also responded to the 2022 survey, which was conducted in September–October 2022. All the participants provided a web-based informed consent before responding to the online self-report questionnaire.

### 2.2. Breast Cancer Screening Programs in Japan

In Japan, breast cancer screening is advised for women aged 40 years and above every two years, with no specified upper age limit [25,26]. Beyond the population-based screening available at municipal units, the Japanese government also endorses opportunistic screenings. These can be availed through employer-provided insurance or can be personally funded by the individuals [25,26]. Under the provisions of the Industrial Safety and Health Act, Japanese employers are mandated to offer annual health check-ups to their permanent staff and those on contracts exceeding one year [27]. Although mammography is the suggested screening method, alternative modalities like clinical breast examinations and breast ultrasonography are available subject to the decisions of the respective municipalities and/or medical institutions coordinating the programs.

### 2.3. Outcome Variable

The primary outcome pertains to participation in breast cancer screening over the past two years, as indicated in the 2022 JACSIS survey. We believe this period aligns with the initial two years of the COVID-19 pandemic. The participants were asked if they had partaken in breast cancer screening methods, including mammography or breast ultrasound, within the preceding 2 years. Responses were restricted to binary options: yes or no.

### 2.4. Exposure Variable

We adopted the following factors as explanatory variables based on previous studies [28,29,30]. We examined sociodemographic factors, health-related behavior traits, and personal behavior characteristics. The sociodemographic variables we considered were age (segmented into 40s, 50s, 60s, and 70s); marital status (divided into married, never married, separated, and divorced) (in the survey, ‘married’ includes both formally registered and cohabitating couples as well as same-sex partners); living arrangement (living alone or in cohabitation); educational attainment (grouped as university or above, junior high/high school, and vocational school/junior/technical college); household annual income (below JPY 3 million and above JPY 3 million); and employment status (employed versus unemployed). For household annual income, JPY3 million was selected as it is considered to be a relative poverty threshold [31].

This study considered health-related behavior characteristics, including pre-COVID-19 breast cancer screening uptake, COVID-19 vaccination status, history of COVID-19 infection, comorbidity, the presence of a family doctor, compliance to COVID-19 preventive measures, and fear of COVID-19. We used the 2021 survey for the pre-COVID-19 breast cancer screening uptake and we used the 2022 survey for other variables. Pre-COVID-19 breast cancer screening uptake was determined as yes or no. COVID-19 vaccination status was categorized into fully vaccinated, partially vaccinated (those who had taken fewer doses than the government’s recommendation), unvaccinated due to health reasons, and unvaccinated for other reasons. Comorbidity was defined as having any of the following chronic diseases, CVDs, diabetes, asthma, stroke, COPD, CKD, hepatitis, or mental disorders, with responses categorized as ‘yes’ or ‘no’. Whether a respondent had a family doctor available for daily consultation determined the presence of a family doctor.

To assess compliance with COVID-19 preventive measures, we used the 14 items related to lack of compliance regarding preventive measure with four options on a scale of 1–4 (1 for always complied and 4 for no compliance at all). A Cronbach’s alpha analysis showed a high level of internal consistency among the 14 scale items (α ≈ 0.8561), indicating that they reliably measure the same underlying construct. We then generated an aggregate ‘lack of compliance’ variable which calculates the row-wise mean across these 14 individual variables, effectively capturing the average level of compliance for each participant. Subsequently, the mean and standard deviation (SD) for each of these 14 individual preventive measures were computed.

The level of anxiety concerning COVID-19, referred to in this study as the ‘Fear of COVID-19 score’, was determined using the Japanese version of the Fear of COVID-19 Scale (FCV-19S) [32], which is consistent with the approach in a previous paper [33]. Scores were classified as low for score of less than 21 points and high for scores above 21 points. We also examined personal behaviors, encompassing smoking status, alcohol consumption, and compliance to COVID-19 preventive measures. Smoking status was segmented into non-smoker, former-user, occasional user, and current user. Alcohol consumption was categorized as ‘never’ for non-drinkers, ‘ever’ for those who have consumed alcohol but not regularly, and ‘current’ for regular drinkers.

### 2.5. Data Analysis

The participant selection process is depicted in Figure 1. For this study, we considered participants who responded to both the 2021 and 2022 surveys. We consolidated the data to form panel data, drawing from 31,000 respondents in 2021 and 32,000 in 2022. This methodology enabled us to investigate the correlation between COVID-19 vaccination and breast cancer screening uptake pre- and post-pandemic. After omitting invalid responses, our sample size was 19,482 participants. Upon further exclusion of individuals below 40 or above 74 years of age, male respondents, and those with a history of any cancer type, our final analytical sample comprised 6110 participants.

First, a descriptive analysis was conducted to give a detailed overview of the dataset, highlighting the primary trends identified among the study participants. The breast cancer screening uptake for the entire sample was calculated, and its distribution was assessed concerning different exposure variables. Following this, an unpaired t-test was employed to explore the relationship between adherence to preventive measures (treated as a continuous variable) and the uptake of breast cancer screening.

Secondly, we formulated a multivariable Poisson regression model to analyze breast cancer screening uptake, incorporating all exposure variables as covariates. Because the prevalence of the outcome was more than 10%, we consider Poisson regression models to be suitable to calculate the Incidence Rate Ratio (IRR) and 95% confidence intervals (CIs) for breast cancer screening uptake [34,35]. Our primary objective was to explore any potential correlation between COVID-19 vaccination and breast cancer screening uptake. To identify multicollinearity among the independent variables, we utilized variance inflation factors (VIFs).

Furthermore, an additional sensitivity analysis was initiated to investigate the dose–response relationship between the mRNA vaccine and post-pandemic breast cancer screening uptake. The mRNA vaccine was chosen due to its prevalent use in Japan and its standard two-dose regimen, which differs from other significant vaccines produced by AstraZeneca and Johnson & Johnson. All statistical analyses, including descriptive statistics, were executed using Stata 14 (StataCorp LP, College Station, TX, USA).

## 3. Results

Table 1 presents a comprehensive breakdown of participant demographics and the breast cancer screening uptake for both the entire cohort and specific subgroups based on the variables considered in this study. Out of 6110 participants, 2870 (46.9%) indicated that they had a breast cancer screening post-COVID-19 pandemic. The age brackets of 40s and 50s registered the highest screening uptakes, with 50.3% and 50.1%, respectively. Those cohabiting showed a marginally increased screening uptake of 47.6% compared to 44.3% for those living alone. Participants who were married reported the most significant uptake: 49%. Those who are gainfully employed displayed a higher screening uptake (51.2%) compared to the unemployed, who had a rate of 42.4%. Additionally, 5421 women (88.7%) in the study were fully vaccinated against COVID-19, and among them 93.6% opted for a breast cancer screening post-pandemic. Regarding prior screening behavior, 2337 (38.2%) had been consistent participants before the pandemic and showed a high screening uptake of 85.4% after the pandemic. Furthermore, 8.9% of the women in our study had a history of COVID-19 infection. The screening participation rates were similar between those with a history of the infection and those without, at 48.5% and 46.8%, respectively.

Table 2 details the participants’ compliance to various COVID-19 preventive measures and their association with breast cancer screening uptake. Participants in the study reported a total average compliance score of 1.59 (standard deviation (SD) = 0.47). The group that participated in post-pandemic breast cancer screening had a slightly lower average score of 1.57 (SD = 0.45) compared to the group that did not participate, with an average score of 1.62 (SD = 0.48). This difference was statistically significant with a *p*-value of 0.0001. Additionally, after applying the Bonferroni correction to account for multiple comparisons across 14 variables, the significance threshold was adjusted to an alpha level of 0.0035. The observed difference in compliance scores between the groups retains its statistical significance.

Table 3 showcases a multivariable Poisson regression model detailing the uptake of breast cancer screening post-COVID-19 pandemic. Individuals who remained unvaccinated due to health concerns (incidence rate ratio (IRR) = 0.47, 95% confidence interval (CI) 0.29–0.77, *p* = 0.003) and for other unspecified reasons (IRR = 0.73, 95% CI 0.62–0.86, *p* < 0.001) were significantly less inclined to opt for screening when compared to their fully vaccinated counterparts. Regarding other factors, individuals in their 60s (IRR = 0.88, 95% CI 0.79–0.98, *p* = 0.027) and 70s (IRR = 0.84, 95% CI 0.73–0.96, *p* = 0.014) were less inclined to undergo screening than the reference group in their 40s. Those who consistently underwent breast screening prior to the pandemic were much more likely to continue post-pandemic (IRR = 3.47, 95% CI 3.19–3.76, *p* < 0.001). Individuals with a family doctor showed a higher likelihood of participating in screening (IRR = 1.12, 95% CI 1.03–1.21, *p* = 0.003). Notably, those who had never been married were considerably less likely to be screened (IRR = 0.71, 95% CI 0.57–0.89, *p* = 0.003). Participants with the lowest educational attainment were also less prone to undergo screening (IRR = 0.83, 95% CI 0.7–0.99, *p* = 0.047). Moreover, individuals earning less than JPY 3 million annually showed a reduced likelihood of screening (IRR = 0.83, 95% CI 0.71–0.97, *p* = 0.019). Intriguingly, those with a heightened fear of COVID-19 were more likely to be screened (IRR = 1.2, 95% CI 1.04–1.39, *p* = 0.012).

Table 4 explores the potential dose-dependent relationship between mRNA vaccine doses and breast cancer screening uptake following the pandemic. Receiving two, three, or four doses of mRNA vaccines was linked to a notable rise in breast cancer screening uptake (two doses IRR = 1.37, 95% CI 1.14–1.64, *p* = 0.001; three doses IRR = 1.31, 95% CI 1.13–1.52, *p* < 0.001; four doses IRR = 1.46, 95% CI 1.24–1.70, *p* < 0.001). However, the data did not indicate a clear dose-dependent trend, as a higher number of vaccine doses was not consistently correlated with increased rates of breast cancer screening uptake.

## 4. Discussion

Our findings reveal that the breast cancer screening uptake during the first two years in the aftermath of the COVID-19 pandemic was 46.9%, which was no lower than the pre-pandemic uptake rate of 38.2% [36]. We also found that there was an association between vaccination against COVID-19 and breast cancer screening uptake rates, though there were no apparent dose relationships between the number of vaccinations and the screening uptake rate. Furthermore, our regression analysis revealed that various factors remained significant factors, including age and previous breast cancer screening history.

Our results suggest that, in contrast to prior studies indicating a decrease in screening rates following the pandemic, the influence of COVID-19 on breast cancer screening in our research was relatively minimal. In fact, screening rates were not only sustained but even exceeded pre-pandemic figures. One plausible explanation for this unexpected trend is the two-year span we considered for our outcome measurement. Although there might have been a noticeable drop in screenings right after the outbreak began, the longer view suggests a strong resurgence in the subsequent months. Notably, in the initial phase of the pandemic, Japan experienced a 10 to 30 percent dip in breast cancer screening uptake, mirroring patterns observed globally [13,14,15,37]. However, our findings underscore a significant bounce-back within two years post-pandemic. This contrasts with data from the US, where, despite a rise in screening rates, they have yet to reach 90% of their pre-pandemic levels [36]. Factors potentially bolstering this resilience include the distinct health consciousness inherent to the Japanese populace and the buttressing effect of national policies like universal healthcare and community-focused medical services [38,39]. It is also possible that the promotion of health behaviors through vaccination further amplified this resilience [20], synergizing with the existing intentions and actions of the public.

Our findings also underscore a significant relationship between COVID-19 vaccination rates and breast cancer screening uptake. Indeed, a staggering 88.7% of participants were fully vaccinated, and this cohort made up 93.6% of those who underwent post-pandemic breast cancer screening. Given that our study is cross-sectional, while it identifies and clarifies correlations, it cannot establish a direct cause-and-effect relationship. However, it is plausible that individuals who are inclined to adopt one preventive behavior, such as getting vaccinated, may also lean towards other preventive measures like breast cancer screening [40,41]. It remains possible that an external factor influenced both the decision to get vaccinated and to pursue breast cancer screening. Further in-depth investigations, including qualitative studies, will be crucial to understand the nuances underpinning this observed relationship.

On the other hand, while our results highlighted an association between vaccination against COVID-19 and breast cancer screening uptake rates, the relationship was not strictly linear in terms of the number of vaccine doses received and the screening uptake rate. Therefore, ensuring a robust vaccination infrastructure was a pivotal first step, and it is deemed that interventions focusing on the psychological and environmental aspects played a significant role in influencing their health behavior decisions [42,43]. The absence of a clear dose–response relationship calls for further investigation into factors like vaccine perceptions, health literacy, trust in healthcare institutions, and personal pandemic experiences, which might underpin the behavioral, psychological, or social mechanisms influencing individual decisions on preventive health measures [44].

Moreover, a crucial observation from this study was the varying uptake rates among different age groups. Those in their 40s and 50s had the highest uptake rates, at 50.3% and 50.1%, respectively. In contrast, those in their 60s and 70s were notably less likely to participate in post-pandemic screenings, as supported by the IRR values presented in Table 3 and Table 4. Beyond age, factors like the presence or absence of a family physician, marital status, education level, employment status, and fear related to COVID-19 also played significant roles in screening uptake. Such differences underscore the importance of age-specific interventions, perhaps incorporating behavioral nudges like reminders or simplified processes to enhance participation, especially in these vulnerable age groups [45].

Also, there was an association between the pre-pandemic uptake and that since the inception of the COVID-19 pandemic. Indeed, the participants who had regularly undergone screenings before COVID-19 demonstrated an 85.3% post-pandemic uptake rate. The current observation aligns with findings from our previous research on the long-term trends of breast cancer screening rates in areas affected by the Great East Japan Earthquake [46]. In the study, one of the factors identified for not undergoing breast cancer screening post-disaster was the absence of a screening history before the earthquake [46]. Given these consistent findings across different crises, it becomes imperative to establish early and regular screening habits in individuals, as such habits prove resilient even in the face of significant adversities. Outreach and education efforts should be prioritized to ensure that more people initiate breast cancer screening early on, thereby securing their continued participation regardless of unforeseen challenges [47].

Our study offers valuable perspectives, but it comes with several caveats. Firstly, the data are self-reported, which might introduce certain biases. Secondly, due to its observational design, we cannot establish clear causal relationships. Notably, our web survey did not specify the chronological order of vaccinations and screenings, meaning screenings might have occurred before vaccinations. This temporal ambiguity should be taken into account when interpreting the observed associations between these variables. Thirdly, the findings were derived from online surveys and primarily represent individuals in Japan with internet access. Thus, the results might not reflect the wider Japanese population or other global communities. Additionally, the reliance on self-reported data may introduce recall bias. The 2021 survey asked participants if they had undergone screening in the past two years, which may have included screenings conducted in 2020, after the COVID-19 outbreak, but before any lockdown measures. This overlap could affect the accuracy of classifying screening behavior as exclusively pre-pandemic. Finally, we did not factor in any initiatives or strategies deployed by healthcare institutions or local authorities to boost breast cancer screening rates.

Despite these limitations, our findings underscore the heterogeneity in the impact of the pandemic on screening rates. This suggests the necessity to tailor interventions to address barriers faced by specific subgroups, particularly reaffirming the significance of past health behaviors as predictors of future preventive actions. Consequently, future public health initiatives should recognize these interdependencies and devise comprehensive strategies to ensure the utilization of health services, especially in crisis situations.

## 5. Conclusions

Our research indicates that breast cancer screening rates in Japan did not decline within the two-year period following the onset of the COVID-19 pandemic. A regular dose of COVID-19 vaccinations was associated with this increased uptake. Further, other notable associated factors included uptake of breast cancer screening before the pandemic and young age. Leveraging these insights is crucial to devise strategies that can effectively address health disparities, aiming for improved public health outcomes in the post-pandemic landscape.

## Figures and Tables

**Figure 1 cancers-16-01783-f001:**
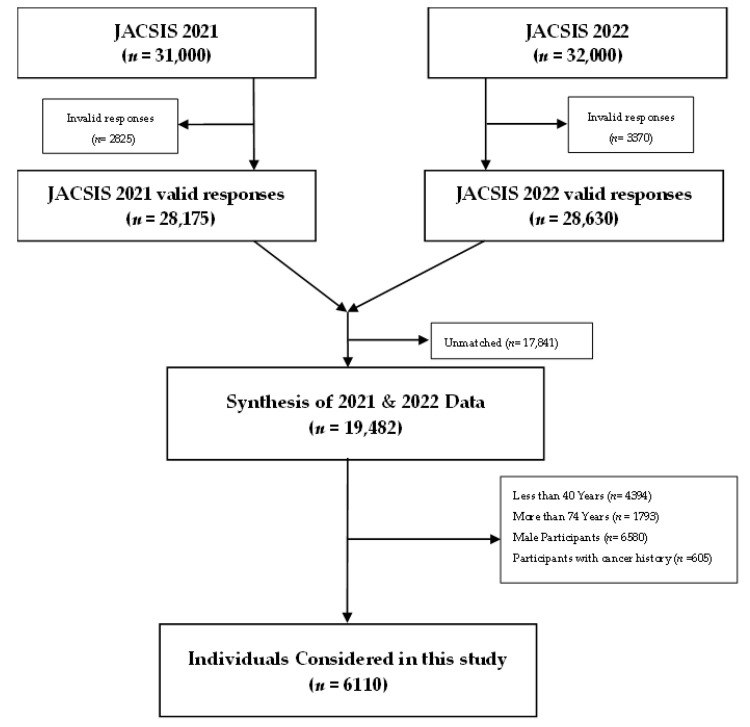
Selection process of the participants in this study.

**Table 1 cancers-16-01783-t001:** Demographic distribution and breast cancer screening uptake among participants post-cOVID-19 pandemic.

Demographic Characteristics	Number and Percentage of All Participants (N = 6110)	Post-Pandemic Breast Cancer Screening Participation Rate (46.9%)	Number and Percentage of Participants Screened Post-Pandemic (N = 2870)
**Age Category**			
40s	1770 (29.0%)	50.3%	891 (31.0%)
50s	1674 (27.4%)	50.2%	840 (29.3%)
60s	1703 (27.9%)	44.8%	763 (26.6%)
70s	963 (15.8%)	39.0%	376 (13.1%)
**Living Status**			
Cohabitating	4997 (81.8%)	47.6%	2377 (82.8%)
Living Alone	1113 (18.2%)	44.3%	493 (17.2%)
**Marital Status**			
Married	4060 (66.4%)	49.0%	1990 (69.3%)
Never Married	1058 (17.3%)	42.0%	444 (15.5%)
Separated	316 (5.2%)	38.0%	120 (4.2%)
Divorce	676 (11.1%)	46.7%	316 (11.0%)
**Education**			
Junior High/High School	2113 (34.6%)	41.9%	886 (30.9%)
Vocational School/Junior College/Technical College	2182 (35.7%)	47.5%	1037 (36.1%)
University or above	1815 (29.7%)	52.2%	947 (33.0%)
**Employment Status**			
Employed	3192 (52.2%)	51.2%	1634 (56.9%)
Unemployed	2918 (47.8%)	42.4%	1236 (43.1%)
**Annual Income**			
Less than 3 million	1176 (19.2%)	38.0%	447 (15.6%)
Greater than 3 million	3370 (55.2%)	51.1%	1723 (60.0%)
Not answered	1564 (25.6%)	44.8%	700 (24.4%)
**Drinking Habit**			
Never	887 (14.5%)	45.1%	400 (13.9%)
Ever	2455 (40.2%)	46.3%	1137 (39.6%)
Current	2768 (45.3%)	48.2%	1333 (46.4%)
**Smoking Status**			
Never user	4062 (66.5%)	48.3%	1960 (68.3%)
Former user	682 (11.2%)	43.4%	296 (10.3%)
Occasional User	753 (12.3%)	49.0%	369 (12.9%)
Current User	613 (10.0%)	40.0%	245 (8.5%)
**Comorbidity**			
No	3309 (54.2%)	48.1%	1593 (55.5%)
Yes	2801 (45.8%)	45.6%	1277 (44.5%)
**Vaccination Status**			
Fully Vaccinated	5421 (88.7%)	49.6%	2687 (93.6%)
Partially Vaccinated	20 (0.3%)	35.0%	7 (0.2%)
Unvaccinated Health Reasons	93 (1.5%)	17.2%	16 (0.6%)
Unvaccinated Other Reasons	576 (9.4%)	27.8%	160 (5.6%)
**History of COVID-19 Infection**			
No	5566 (91.1%)	46.8	2606 (90.8%)
Yes	544(8.9)	48.5	264 (9.2%)
**Breast Screening Uptake before COVID-19**			
Irregular and No participation	3773 (61.8%)	23.2%	875 (30.5%)
Regular Participation	2337 (38.2%)	85.4%	1995 (69.5%)
**COVID-19 Fear Score**			
Less than 21 points	4379 (71.7%)	46.4%	2032 (70.8%)
21 points or more	1731 (28.3%)	48.4%	838 (29.2%)
**Presence of Family Doctor**			
No	2830 (46.3%)	41.9%	1185 (41.3%)
Yes	3280 (53.7%)	51.4%	1685 (58.7%)
**Compliance to COVID-19 Preventive Measures**			
Mean ± SD	1.600 ± 0.467		1.575 ± 0.447

Note: SD = standard deviation. Compliance to COVID-19 preventive measures is presented as a continuous variable and reported using the mean and standard deviation.

**Table 2 cancers-16-01783-t002:** Compliance to preventive measures and their associations with breast cancer screening uptake after COVID-19 pandemic (N = 6110).

	Mean (SD)	Non-uptake	Uptake	*p*-Value ^†^
**Total average**	1.59 (0.47)	1.62 (0.48)	1.57 (0.45)	0.0001
**Disinfecting hands with rubbing alcohol**	1.38 (0.65)	1.41 (0.69)	1.32 (0.59)	<0.0001
**Washing hands for 15 s or longer with soap**	1.47 (0.71)	1.50 (0.75)	1.43 (0.68)	0.0002
**Gargle after returning home**	1.87 (1.02)	1.93 (1.05)	1.80 (0.99)	<0.0001
**Practice cough etiquette**	1.22 (0.61)	1.25 (0.65)	1.18 (0.61)	<0.0001
**Avoid touching eyes, nose, and mouth with unwashed hands**	1.60 (0.79)	1.64 (0.82)	1.57 (0.76)	0.0006
**Disinfect objects that are easily touched by people, such as doorknobs**	2.36 (0.99)	2.44 (1.00)	2.26 (0.98)	<0.0001
**Open the window to ventilate the room**	1.51 (0.72)	1.53 (0.75)	1.49 (0.68)	0.0412
**Wearing a mask when there are people around**	1.06 (0.33)	1.06 (0.36)	1.05 (0.31)	0.1916
**Refrain from traveling**	1.55 (0.87)	1.52 (0.87)	1.60 (0.88)	0.0002
**Refrain from unnecessary and non-urgent outings and business trips**	1.73 (0.89)	1.71 (0.89)	1.76 (0.88)	0.0205
**Avoid talking or vocalizing at a short distance (within 1 m)**	1.76 (0.83)	1.78 (0.84)	1.75 (0.82)	0.2352
**I tried to take social distance (at least 2 m away from people)**	1.73 (0.79)	1.75 (0.82)	1.71 (0.79)	0.0277
**Avoided meeting with people thought to be at high risk of infection**	1.56 (0.85)	1.59 (0.88)	1.53 (0.80)	0.0046
**Avoid going to crowded places**	1.56 (0.75)	1.56 (0.77)	1.57 (0.73)	0.7928

^†^ Unpaired *t*-test to evaluate potential association between compliance to preventive measure and breast cancer screening uptake after the COVID-19 pandemic.

**Table 3 cancers-16-01783-t003:** Multivariable Poisson regression model for breast cancer screening uptake after COVID-19 pandemic (N = 6110).

	Total Population (N = 6110)		
Demographic Characteristics	IRR *	95% CI ^†^	*p*-Value
**Age Category**			
40s	Reference		
50s	0.93	(0.84–1.03)	0.172
60s	0.88	(0.79–0.98)	0.027
70s	0.84	(0.73–0.96)	0.014
**Living Status**			
Living Alone	Reference		
Cohabitating	0.93	(0.82–1.06)	0.335
**Marital Status**			
Married	Reference		
Never Married	0.88	(0.77–1.00)	0.058
Separated	0.95	(0.78–1.17)	0.686
Divorced	0.99	(0.87–1.14)	0.974
**Education**			
University or above	Reference		
Junior High/High School	0.93	(0.85–1.02)	0.170
Vocational School/Junior College/Technical College	0.98	(0.90–1.08)	0.814
**Employment Status**			
Employed	Reference		
Unemployed	0.94	(0.87–1.03)	0.227
**Annual Income**			
Greater than 3 million	Reference		
Less than 3 million	0.91	(0.81–1.02)	0.118
Not answered	0.93	(0.85–1.02)	0.137
**Smoking Status**			
Never user	Reference		
Former user	0.97	(0.86–1.10)	0.733
Occasional User	1.02	(0.91–1.14)	0.707
Current User	0.94	(0.82–1.08)	0.447
**Drinking Habit**			
Never	Reference		
Ever	1.00	(0.89–1.12)	0.942
Current	0.99	(0.88–1.11)	0.944
**Comorbidity**			
No	Reference		
Yes	0.96	(0.89–1.04)	0.377
**Vaccination Status**			
Fully Vaccinated	Reference		
Partially Vaccinated	1.01	(0.48–2.13)	0.976
Unvaccinated Health reason	0.47	(0.29–0.77)	0.003
Unvaccinated Other Reasons	0.73	(0.62–0.86)	<0.001
**History of COVID-19 Infection**			
No	Reference		
Yes	0.96	(0.89–1.09)	0.580
**Breast Screening Uptake before COVID-19 pandemic**			
Irregular, No participation	Reference		
Regular Participation	3.47	(3.19–3.76)	<0.001
**Fear of COVID-19 Score**			
Less than 21 points	Reference		
21 points or more	1.05	(0.97–1.15)	0.174
**Presence of Family Doctor**			
No	Reference		
Yes	1.12	(1.03–1.21)	0.003
**Compliance to COVID-19 Preventive Measure**	0.99	(0.91–1.08)	0.902

Note: Compliance to COVID-19 preventive measures is analyzed as a continuous variable. * IRR; incidence rate ratio, CI ^†^; confidence interval.

**Table 4 cancers-16-01783-t004:** Dose relationship in participants with mRNA vaccine and screening uptake after COVID-19 pandemic (N = 6110).

Demographic Characteristics	Incidence Rate Ratio	95% Confidence Interval	*p*-Value
**Age Category**			
40s	Reference		
50s	0.93	(0.85–1.03)	0.185
60s	0.84	(0.74–0.95)	0.005
70s	0.78	(0.67–0.91)	0.002
**Living Status**			
Living Alone	Reference		
Cohabitating	0.93	(0.82–1.06)	0.331
**Marital Status**			
Married	Reference		
Never Married	0.88	(0.77–1.00)	0.059
Separated	0.95	(0.78–1.16)	0.649
Divorced	0.99	(0.86–1.14)	0.960
**Education**			
University or above	Reference		
Junior High/High School	0.93	(0.85–1.02)	0.169
Vocational School/Junior College/Technical College	0.98	(0.90–1.07)	0.735
**Employment Status**			
Employed	Reference		
Unemployed	0.94	(0.86–1.03)	0.208
**Annual Income**			
Greater than 3 million	Reference		
Less than 3 million	0.91	(0.81–1.02)	0.135
Not answered	0.93	(0.85–1.02)	0.172
**Smoking Status**			
Never user	Reference		
Former-user	0.98	(0.86–1.11)	0.770
Occasional User	1.02	(0.91–1.14)	0.698
Current User	0.94	(0.82–1.08)	0.453
**Drinking Habit**			
Never	Reference		
Ever	1.00	(0.89–1.12)	0.970
Current	0.99	(0.88–1.11)	0.950
**Comorbidity**			
No	Reference		
Yes	0.95	(0.88–1.03)	0.281
**History of COVID-19 Infection**			
No	0.96	(0.84–1.10)	0.542
Yes			
**Breast Screening Uptake before COVID-19**			
Irregular, No participation	Reference		
Regular Participation	3.47	(3.19–3.76)	<0.001
**Fear of COVID-19 Score**			
Less than 21 points	Reference		
21 points or more	1.06	(0.98–1.15)	0.141
**Presence of Family Doctor**			
No	Reference		
Yes	1.11	(1.03–1.21)	0.005
**mRNA Doses**			
Zero dose	Reference		
One dose	1.41	(0.66–3.00)	0.372
Two doses	1.37	(1.14–1.64)	0.001
Three doses	1.31	(1.13–1.52)	<0.001
Four doses	1.46	(1.24–1.70)	<0.001
**Compliance to COVID-19 Preventive Measure**	1.00	(0.92–1.09)	0.934

Note: Compliance to COVID-19 preventive measures is analyzed as a continuous variable.

## Data Availability

The datasets generated and analyzed during the current study are not publicly available but are available from the corresponding author on reasonable request.

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
