# Peer review of "Two-Year-Span Breast Cancer Screening Uptake in Japan after the COVID-19 Pandemic and Its Association with the COVID-19 Vaccination"

_cancers, 2024, doi:10.3390/cancers16091783_

Round 1

Reviewer 1 Report

Comments and Suggestions for Authors

Minor revisions:

·         Line 94: with no family history of cancer/ breast cancer?

·         Table 1: N for all participants is wrong (6110 instead of 6100)

·         Categorization of Marital Status: Would it not make sense to add a category “in a partnership” instead of having those not married but in a partnership included in the group of single persons (never married). Those with a partner might have similar associations as those who are married.

·         Table 3: How is regular and irregular participating in breast screening uptake before COVID-19 pandemic defined? In line 131/132, it says that Pre COVID-19 breast cancer screening uptake was determined as yes or no.

·         Recall bias should be added as limitation

Major revisions:

·         Line 30-31: This could be also due to a bias, that women who get vaccinated also care more about screening programs

·         Line 115 & 175: Poisson regression is used for a binary outcome. Or is there a misunderstanding in line 115, that you count the number of breast cancer screenings during the last 2 years? Please specify with more details or explain why Poisson regression is used for a binary outcome (it is possible but not that common).

·         Lines 190-200: In the text, you compare with the percentages the participants with breast cancer screening against all participants. This might not be clear to the reader as in the table, you have column percentage and in the text you use the row percentage

·         As you include the same participants at two timepoints, would It not make sense to perform a longitudinal analysis using mixed effect models?

Comments on the Quality of English Language

I just noticed two minor issues in line 33 (remove "a") and in line 79 (remove one "yet").

Author Response

Thank you for your thorough review and valuable feedback on our manuscript. I have carefully considered your comments and made the necessary adjustments as suggested.

Minor revisions:

  • Line 94: with no family history of cancer/ breast cancer?
  • This is a mistake. Participants with a history of cancer were excluded. The survey had no questions regarding a family history of cancer. Thank you for pointing this out. The sentence has been updated on line 94.
  • Table 1: N for all participants is wrong (6110 instead of 6100)
  • The correction has been made
  • Categorization of Marital Status: Would it not make sense to add a category “in a partnership” instead of having those not married but in a partnership included in the group of single persons (never married). Those with a partner might have similar associations as those who are married.
  • In the survey, "been married" also includes those who live together without formal marriage, as well as same-sex partners. The sentence has been updated in Line 121.
  • Table 3: How is regular and irregular participation in breast screening uptake before COVID-19 pandemic defined? In line 131/132, it says that Pre COVID-19 breast cancer screening uptake was determined as yes or no.
  • The 2021 survey asked participants whether they had undergone breast cancer screening in the last two years. This was considered pre-COVID. Explanations regarding this point and descriptions have been added to the limitations in lines 328-331.
  • Recall bias should be added as a limitation.
  • Recall bias has been added as a limitation in line 327. Thank you for bringing this to our attention.   Major revisions:
  • Line 30-31: This could be also due to a bias, that women who get vaccinated also care more about screening programs
  • Based on your suggestion, we have revised the statement to better reflect the expected behaviour among health-conscious individuals who are also likely to participate in vaccination programs. The statement has been updated in lines 31-32.
  • Line 115 & 175: Poisson regression is used for a binary outcome. Or is there a misunderstanding in line 115, that you count the number of breast cancer screenings during the last 2 years? Please specify with more details or explain why Poisson regression is used for a binary outcome (it is possible but not that common).
  • We used Poisson regression models to calculate the Incidence Rate Ratios (IRRs) and 95% Confidence Intervals (CIs) for screening uptake because the outcome prevalence was more than 10%. This approach aligns with methodologies used in previous research (McNutt et al., 2003 & Zou, 2004). We have updated the Methods section of our manuscript to include an explanation and appropriate references supporting our choice.
  • Lines 190-200: In the text, you compare the percentages of the participants with breast cancer screening against all participants. This might not be clear to the reader as in the table, you have column percentage and in the text, you use the row percentage.
  • The table and text results have been updated for clarity.
  • As you include the same participants at two time points, would It not make sense to perform a longitudinal analysis using mixed-effect models?
  • We focused on assessing general trends in breast cancer screening rates rather than individual changes over time align with our study's objective.

  • The typing errors in lines 33 and 79 have been corrected.

Many thanks for your effort, we appreciate your feedback.

Reviewer 2 Report

Comments and Suggestions for Authors

This is an interesting enquiry into how COVID-19 pandemic influenced (or not) medical interventions not associated with it directly, namely breast cancer screening. We need such sort of information to adjust functioning our health systems in the situation of other pandemics. However, I doubt that the core hypothesis of the article may be easily justified (I mean the relationship between COVID-19 vaccination and breast cancer screening rate).

What a reader may be interested in, is:

- what was the screening rate during the pandemic (not before and in the aftermath)? How coronavirus-related restrictions influenced the regime of medical examinations of breast cancer?

- why did the authors ever advance a hypothesis that COVID-19 vaccination rate may be correlated with breast cancer examination? This may seem strange at the first sight. Perhaps, they have to explain their motives more clearly in Introduction. Their suggestion in Discussion (lines 280-283) is a commonplace and cannot ground their hypothesis. Of course, any person accustomed to watch his/her health status, will observe healthcare procedures more closely than a person who does not care. It is a priori right and it does not require any special intervention, nor study.

Another explanation is necessary.

- why did the authors not include those women who got infected with SARS-CoV2 and had the disease, in their statistics? Having the disease is natural vaccination that does not require any medical vaccination afterwards during at least a six-month period. It would be good to know the behaviour of this group in regard to breast cancer screening.

All in all, the authors convey interesting findings but their explanation is loose. I recommend addressing the point that are summarised above, for their paper to become more consistent.

Author Response

Thank you for your thorough review and valuable feedback on our manuscript. I have carefully considered your comments and made the necessary adjustments as suggested.

  • what was the screening rate during the pandemic (not before and in the aftermath)? How coronavirus-related restrictions influenced the regime of medical examinations of breast cancer?

Thank you for highlighting the need to understand the impact of lockdowns on screening rates during the pandemic. Previously, one of our co-authors reported the disruptions in breast cancer screenings in Japan due to COVID-19. Here, we focused on recovery in the post-pandemic period.

  • - why did the authors ever advance a hypothesis that COVID-19 vaccination rate may be correlated with breast cancer examination? This may seem strange at the first sight. Perhaps, they have to explain their motives more clearly in Introduction. Their suggestion in Discussion (lines 280-283) is a commonplace and cannot ground their hypothesis. Of course, any person accustomed to watch his/her health status, will observe healthcare procedures more closely than a person who does not care. It is a priori right and it does not require any special intervention, nor study. Another explanation is necessary.

While it’s commonplace that Individuals who participate in one preventive health measure are often inclined to adopt others, driven by many factors, these factors can be heightened during a pandemic. By hypothesizing a link between COVID-19 vaccination uptake and breast cancer screening, our study aims to explore these correlations in the unique context of a pandemic, adding to existing knowledge to inform public health strategies. It's worth noting that Japan has one of the best vaccination coverages for COVID-19, but cancer screening rates are lower in the country compared to other countries with a relatively similar health profile. Public health campaigns could integrate messaging and interventions that promote vaccinations and cancer screenings simultaneously, leveraging the existing inclination of this population segment towards preventive measures. We've included additional information in the introduction (lines 77-83) to clarify the rationale behind our hypothesis.

  • why did the authors not include those women who got infected with SARS-CoV2 and had the disease, in their statistics? Having the disease is natural vaccination that does not require any medical vaccination afterwards during at least a six-month period. It would be good to know the behaviour of this group in regard to breast cancer screening.

Thank you for your suggestion regarding the inclusion of women who had contracted SARS-CoV-2 in our study. We added a history of COVID-19 infection as a variable in our analysis.

Many thanks for your effort, we appreciate your feedback.

Round 2

Reviewer 1 Report

Comments and Suggestions for Authors

- Table 4: You added "History of Covid19 Infection": Is "Yes" or "No" the reference?

Otherwise, all comments have been addressed and justified. Therefore, I have no further comments or suggestion!

Author Response

Thank you very much for your time. 

No is the reference group for the history of infection variable. 

Reviewer 2 Report

Comments and Suggestions for Authors

The authors have made satisfactory improvements. The paper may be published. No more comments

Author Response

Thank you very much for your time.